

# Persistent Climate Model Biases in the Atlantic Ocean's Freshwater Transport

René M. van Westen[1] and Henk A. Dijkstra[1]

[1]Institute for Marine and Atmospheric research Utrecht, Department of Physics, Utrecht University, Utrecht, the Netherlands

**Correspondence:** René M. van Westen <r.m.vanwesten@uu.nl>

**Abstract.** The Atlantic Meridional Overturning Circulation (AMOC) is considered to be one of the most dangerous climate tipping elements. From idealised model studies, it is known that the tipping behaviour is caused by a positive salt-advection feedback, which is strongly connected to the Atlantic Ocean's freshwater transport. In earlier model studies, using climate models of the Coupled Model Intercomparison Projects (phase 3 and phase 5), biases in this freshwater transport have been identified. Here, we show that these biases persist in CMIP phase 6 models, as well as in a climate model with an eddying ocean, and provide a more detailed analysis of the origin of the biases. The most important model bias is in the Atlantic Surface Water properties, which arises from deficiencies in the surface freshwater flux over the Indian Ocean. The second largest bias is in the properties in the North Atlantic Deep Water and arises through deficiencies in the freshwater flux over the Atlantic Subpolar Gyre region. Due to the biases, the Atlantic Ocean's freshwater transport is not in agreement with available observations and the strength of the salt advection feedback is underestimated. To better project future AMOC behaviour, an urgent effort is needed to reduce biases in the atmospheric components of current climate models.

## 1 Introduction

The Atlantic Meridional Overturning Circulation (AMOC) plays an important role in global climate because of its meridional transport of heat and salt. The present-day AMOC has a strength of $16 - 19$ Sv ($1$ Sv $= 10^6$ m s$^{-1}$) near $26°$N (Smeed et al., 2018) and effectively transports heat northwards, with a value of 1.5 PW at $26°$N (Johns et al., 2011). The AMOC is considered to be one of the most important tipping elements (Armstrong McKay et al., 2022) and could, under future climate change, collapse to a state with a much weaker strength and corresponding weaker heat transport. It is a dangerous tipping element because, due to an AMOC collapse, large changes in sea surface temperatures, precipitation patterns, sea level and tropical cyclones (McFarlane and Frierson, 2017; Orihuela-Pinto et al., 2022; van Westen et al., 2023) can occur within a few decades.

Although reconstructed time series of the AMOC strength over the historical record appear to indicate a weakening of the AMOC (Caesar et al., 2021), the more recent direct observations indicate no decline in AMOC strength over the past 30 years (Worthington et al., 2021). Both time series of AMOC strength are relatively short and no AMOC collapses have been found. The idea of an AMOC collapse originates from conceptual models (Stommel, 1961; Castellana et al., 2019) and such collapses have been found in Earth System Models of Intermediate Complexity (Rahmstorf et al., 2005; Den Toom et al., 2012). The





transitions in these models are related to the existence of a multi-stable AMOC regime where different equilibrium states exist under the same forcing conditions. Transitions between these states are caused by the salt-advection feedback (Marotzke, 2000; Peltier and Vettoretti, 2014), a positive feedback in which salinity anomalies are amplified through their effect on the AMOC strength and pattern.

As a measure of the salt-advection feedback strength, an indicator was developed (Rahmstorf, 1996; de Vries and Weber, 2005) based on the net Atlantic freshwater transport by the AMOC at 34°S (the southern boundary of the Atlantic Ocean) which we denote here as the $F_{\mathrm{ovS}}$ (Weijer et al., 2019). When $F_{\mathrm{ovS}} < 0$ ($> 0$), the AMOC transports net salinity (fresh) water into the Atlantic Ocean and the salt-advection feedback is positive (negative). Present-day hydrographic observations show negative values of $F_{\mathrm{ovS}} < 0$ (Bryden et al., 2011; Garzoli et al., 2013) and also a recent Lagrangian study of reanalysis data shows the same property (Rousselet et al., 2021). Clearly, most models used in the Coupled Model Intercomparison Projects (CMIP) phase 3 (CMIP3) (Drijfhout et al., 2011) and phase 5 (CMIP5) (Mecking et al., 2017) have $F_{\mathrm{ovS}} > 0$ and hence do not adequately capture the salt-advection feedback.

Here we determine the $F_{\mathrm{ovS}}$ biases in 39 CMIP6 models and a high-resolution (HR) and low-resolution (LR) version of the Community Earth System Model (CESM) and add further analyses on their origin. In section 2 a brief description of the HR-CESM, LR-CESM and CMIP phase 6 (CMIP6) models is provided, together with a description of the freshwater transport analysis. In section 3, we systematically analyse the $F_{\mathrm{ovS}}$ biases in the HR-CESM and LR-CESM models and provide a comparison with the biases in the CMIP6 models. A summary and discussion of the results with the main conclusions are given in the final section 4.

## 2  Climate Model Simulations and Methods

We analysed results from the 500-year long pre-industrial (PI) control simulations for the HR-CESM and LR-CESM as provided by Chang et al. (2020). The LR-CESM has a horizontal resolution of 1° for both the ocean and atmosphere components, while the HR-CESM has a strongly eddying ocean (0.1° horizontal resolution) and resolves tropical cyclones in the atmospheric component (0.25° horizontal resolution). The HR-CESM and LR-CESM have the same 60 non-equidistant vertical layers down to 5,375 m, with the highest vertical resolution near the surface (10 m) and lowest resolution near the bottom (250 m). The HR-CESM has two additional vertical layers below 5,375 m but their effect is very limited as only a few grid cells extend below 5,375 m. Increasing the horizontal ocean resolution to 0.1° strongly improves the global ocean circulation and reduces ocean-related biases (Small et al., 2014; Jüling et al., 2021; van Westen et al., 2020; van Westen and Dijkstra, 2021). Both simulations are initialised from an observed ocean state for temperature and salinity. At model year 250 of the PI control simulation, another simulation was branched off which is forced by historical observations (1850 – 2005) and then followed by the RCP8.5 climate change forcing scenario (2006 – 2100), which we refer to as the Hist/RCP8.5 simulation.

For comparison with the Hist/RCP8.5 (1994 – 2020) CESM simulations, we used the eddy-resolving (1/12°) Copernicus Marine global reanalysis product (1994 – 2020) as 'observations'. For the CMIP6 models we retained the historical (1994 – 2014) followed by SSP5-8.5 (2015 – 2100) forcing scenario, which we refer to as the Hist/SSP5-8.5 simulation. Note that





the forcing scenarios are different between the CESM (Hist/RCP8.5) and CMIP6 scenarios (Hist/SSP5-8.5), but the projected

temperature in 2100 are both high-end scenarios (+3°C – +5°C w.r.t. the pre-industrial period). The monthly-averaged model output from the CESM, reanalysis and CMIP6 is converted to yearly-averaged fields. The analyses here are conducted on these yearly-averaged fields and on their native grid.

The freshwater transport by the overturning component ($F_{\mathrm{ovS}}$) and the azonal (gyre) component ($F_{\mathrm{azS}}$) at 34°S are determined as:

$$
F_{\mathrm{ovS}} = F_{\mathrm{ov}}(y = 34°\mathrm{S}) = -\frac{1}{S_0} \int\limits_{-H}^{0} \left[ \int\limits_{x_W}^{x_E} v^* \,\mathrm{d}x \right] [\langle S \rangle - S_0]\,\mathrm{d}z \tag{1a}
$$

$$
F_{\mathrm{azS}} = F_{\mathrm{az}}(y = 34°\mathrm{S}) = -\frac{1}{S_0} \int\limits_{-H}^{0} \int\limits_{x_W}^{x_E} v' S' \,\mathrm{d}z \tag{1b}
$$

where $S_0 = 35$ g kg$^{-1}$ is a reference salinity. The $v^*$ indicates the baroclinic velocity and is defined as $v^* = v - \hat{v}$, where $v$ is the meridional velocity and $\hat{v}$ the section spatially-averaged meridional velocity. In addition, $\langle S \rangle$ indicates the zonally-averaged salinity and primed quantities ($v'$ and $S'$) are deviations from their respective zonal means (Jüling et al., 2021).

The total freshwater transport at 34°S can be separated into a contribution of four different water masses, i.e., the Atlantic Surface Water (ASW), the Antarctic Intermediate Water (AIW), the North Atlantic Deep Water (NADW) and the Antarctic Bottom Water (ABW). The boundaries for the ASW, AIW and NADW and ABW are determined by first locating the NADW layer. This layer has negative baroclinic meridional velocities and is found around 1,000 – 4,000 m depths. Directly above the NADW, where the meridional velocities become positive, we define the AIW. The AIW is bounded above by the 500 m depth

level and the ASW is defined between the 500 m depth level and the surface. The ABW is located directly below the NADW, where the velocities become positive, and extends down to the bottom. The layer thickness of each of these water masses may vary over time due to changes in the meridional velocity profile. We did not define the water masses based on their $T, S$-related properties as climate change alters these properties.

The AMOC strength is defined as the total meridional mass transport at 26°N over the upper 1,000 m:

$$
\mathrm{AMOC}(y = 26°\mathrm{N}) = \int\limits_{-1000}^{0} \int\limits_{x_W}^{x_E} v \,\mathrm{d}x\mathrm{d}z \tag{2}
$$

This AMOC strength may deviate from the maximum AMOC strength as the maximum varies around 1,000 m depth, but using this metric is then consistent between all climate model simulations and reanalysis. Note that some models also provide the AMOC streamfunction as standard output, but for consistency we determined the AMOC strength as in relation 2.

The trends computed below are derived from a linear least-square fit to the yearly-averaged time series. The significance of

each trend is determined following the procedure outlined in Santer et al. (2000), while taking into account the reduction of degrees of freedom for time series which are not statistically independent. Using the reduced degrees of freedom and the two-sided critical Student-$t$ values, one can determine the significance of having a trend different from zero (the null hypothesis).





# 3 Results

## 3.1 The PI Control Simulations

The values of $F_{\text{ovS}}$ for the PI control CESM simulations are shown in the Figures 1a,b, with the PI control in black and the Hist/RCP8.5 simulation in red. The first 20 model years of the HR-CESM PI control are not available. From the initial observed ocean state, it is striking that the value of $F_{\text{ovS}}$ drifts from negative to positive values within the first 250 model years of the PI control simulations. This drift mainly originates from the ASW and the NADW contributions to $F_{\text{ovS}}$ (back curves in the Figures 2a,b and Figures 2e,f). The azonal (gyre) freshwater transport (upper insets in the Figures 1a,b) and the AIW and ABW

contributions (black curves in Figures 2c,d and Figures 2g,h) show adjustments in the first 50 model years and then remain fairly constant over the remaining simulation period.

The ASW's salinity is related to Agulhas Leakage, which in turn is related to the surface salinity of the Indian Ocean. Both the HR-CESM and LR-CESM show a freshening of the Indian Ocean, which is dominated by the precipitation minus evaporation (P-E) response over the Indian Ocean (see bottom insets in Figures 1a,b). Here we show the P-E trends over model

years 21 – 100; the trends over the first 100 years are shown in Figure A1 for the LR-CESM PI control only. The volume-averaged (0 – 100 m) salinity over the Indian Ocean strongly decreases by 0.3 g kg$^{-1}$ in the first 10 years in the LR-CESM (Figure 1d). For the HR-CESM this is only 0.2 g kg$^{-1}$ in the first 20 years (Figure 1c), where we used the initial value of the LR-CESM for reference. The relatively strong freshening in the LR-CESM over the first 20 years explains why the Indian Ocean's salinity trends are smaller compared to those in the HR-CESM. However, considering the first 100 years for the LR-

CESM, also negative salinity trends over the Indian Ocean are found (Figure A1b). The precipitation changes are the dominant driver in the P-E response (Figure A2) and are related to Intertropical Convergence Zone (ITCZ) biases (Mamalakis et al., 2021). The Indonesian Throughflow also imports (net) fresh water into the Indian Ocean (Figure A3), but this can not solely explain the (strong) freshening of the Indian Ocean in the first 5 years of the LR-CESM. The negative salinity anomalies (w.r.t. initialisation) in the Indian Ocean eventually reach the Agulhas retroflection and through Agulhas Leakage affect the ASW.

This leads to positive freshwater anomalies transported into the Atlantic Ocean (see curves in insets in the Figures 1c,d).

The NADW also contributes to the $F_{\text{ovS}}$ drift (Figures 2e,f). The NADW is part of southward flowing limb of the AMOC and this water mass originates from deep water formation at the higher latitudes in the North Atlantic. After deep water formation, the newly-formed water mass takes about 100 years to reach 34°S (Figure A4). This motion in this water mass is linked to the AMOC strength which is shown in the Figures 3a,b. There is some adjustment in the first 100 model years of the PI control

simulations (AMOC is 0 Sv at initialisation), but thereafter it is in near equilibrium. The adjustment in AMOC strength during the first 100 years results in sea surface temperatures (SSTs), P-E and salinities responses (insets in Figure 3). In the HR-CESM, the AMOC strength overshoots in the first 20 years and then decreases in the following 80 years. A weaker AMOC induces an SST response and these SSTs influence the P-E response mainly through evaporation (Figure A5). This characteristic SST pattern looks similar to the fingerprint suggested as a proxy of the strength of the AMOC in observations and models (Caesar

et al., 2018; van Westen et al., 2023).





**Figure 1.** (a & b): The freshwater transport by the overturning component at $34°S$, $F_{ovS}$, (determined at black section in lower inset) for the a) HR-CESM and b) LR-CESM. The yellow shading indicates observed ranges (Garzoli et al., 2013; Mecking et al., 2017). Insets: The freshwater transport by the azonal (gyre) component ($F_{azS}$, at $34°S$) and P-E trends (PI control, model years $21 - 100$). (c & d): The vertically-averaged ($0 - 100$ m) and spatially-averaged salinity over the Indian Ocean for the c) HR-CESM and d) LR-CESM. Inset: The vertically-averaged ($0 - 100$ m) salinity trends (PI control, model years $21 - 100$), the curves indicate the $35$ g kg$^{-1}$ salinity isolines for model year 21 (black) and model year 100 (red). We did not indicate significant trends for visibility reasons.





**Figure 2.** The freshwater transport at $34°$S, separated for the four different water masses for the HR-CESM (left column) and LR-CESM (right column). The cyan-coloured curve shows reanalysis. The opaque curves show the total freshwater transport $F_{\mathrm{ovS}}$ (see also Figures 1a,b), the yellow shading indicates observed ranges for $F_{\mathrm{ovS}}$ (Garzoli et al., 2013; Mecking et al., 2017). Insets: The spatially-averaged meridional velocity and salinity over the water masses for the HR-CESM and LR-CESM.



Positive P-E trends reduce the surface (upper 100 m) salinities south of Greenland (insets in the Figures 3a,c), but the surface salinity increases elsewhere. The East and West Greenland Current play an important role in deep water formation and more saline currents increase the vertically-averaged (1,000 – 3,000 m) salinity of the Labrador basin, Irminger basin and Iceland basin (Figure 3c). The maximum salinity for the different basins is reached around model year 130 and at the same time the

AMOC has a local maximum in its strength. After model year 130, the salinity over the basins steadily decreases and there is a small decline (-0.5 Sv per century, $p < 0.01$, model years 130 – 500) in AMOC strength. After deep water formation the more saline water flows southward at depth and increases the salinities at lower latitudes as well (inset in Figure 3c).

For the LR-CESM, the AMOC increases in strength over the first 100 model years, whereas it decreases for HR-CESM. Consequently, we find the opposite SST, P-E and upper 100 m salinity responses for both model versions (Figures 3b,d and

A1). The LR-CESM surface salinity trends around Greenland are much stronger (related to the different SST response and stronger evaporation, Figure A5) compared to the HR-CESM. The salinity over the three basins increases and their maximum is reached around model year 65, which is twice as fast compared to the HR-CESM. The AMOC strength in LR-CESM reaches its maximum around model year 65 (the same as it reaches a maximum in HR-CESM) and then decreases in strength by -0.2 Sv per century ($p < 0.01$, model years 130 – 500).

The salinity response near the deep water formation regions is about twice as strong in the LR-CESM compared to the HR-CESM over the first 100 model years. Yet, the NADW freshwater transport (at 34°S, Figures 2e,f) shows a stronger drift (model years 100 – 250) in the HR-CESM (0.038 Sv per century, $p < 0.01$) than the LR-CESM (0.014 Sv per century, $p < 0.01$). The differences in the NADW freshwater transport trends are related to the ventilation rate of the NADW. First, the southward flowing water at depth (formed at the higher latitudes in the North Atlantic) reaches 34°S faster in the HR-CESM than the

LR-CESM, which is presented here as the spatially-averaged age over the NADW (Figure A4g). Second, the relatively young water mass in the NADW has a larger zonal extent in the HR-CESM than in the LR-CESM (compare Figures A4c and A4d). The high horizontal ocean model resolution in the HR-CESM results in much more eddy-induced horizontal mixing (w.r.t. the LR-CESM) which gives rise to a greater ventilation rate of the NADW at 34°S. After model year 250, the NADW freshwater transport slightly declines again (-0.011 Sv per century, $p < 0.01$) in the HR-CESM, which is consistent with the salinity

maxima in the Labrador basin, Irminger basin and Iceland basin that are reached 100 years earlier. Over this later period, the LR-CESM shows a persistent positive NADW trend (0.004 Sv per century, $p < 0.01$) which contributes to the drift in $F_{ovS}$. This indicates that the salinity content of the deeper ocean in the LR-CESM takes a much longer time to adjust than the HR-CESM (see also insets in Figures 2e,f), in particular given that the salinity maxima of the Labrador basin, Irminger basin and Iceland basin are reached around model year 65 for the LR-CESM.

The Atlantic's northern boundary (at 60°N, $F_{ovN}$) also contributes to the freshwater budget of the Atlantic Ocean and the convergence/divergence of freshwater by the overturning circulation is indicated by $\Delta F_{ov} = F_{ovS} - F_{ovN}$ (Dijkstra, 2007; Weijer et al., 2019). For the HR-CESM PI control, $F_{ovN}$ is about -0.03 Sv and its magnitude is smaller than $F_{ovS}$ (Figure 4a), and hence $\Delta F_{ov} \approx F_{ovS}$. For the LR-CESM PI control, $F_{ovN}$ contributes quite some more to $\Delta F_{ov}$ (Figure 4b). As a result, the values of $\Delta F_{ov}$ are fairly similar for the HR-CESM and LR-CESM between model years 200 – 500.



**Figure 3.** (a & b): The AMOC strength at 1,000 m and 26°N (determined at black section in left inset) for the a) HR-CESM and b) LR-CESM. The yellow shading indicates observed ranges (Smeed et al., 2018; Worthington et al., 2021) for the AMOC strength at 1,000 m and 26°N. Insets: The SST and P-E trends (PI control, model years 21 – 100). (c & d): The vertically-averaged (1,000 – 3,000 m) and spatially-averaged salinity over the Labrador basin, Irminger basin and Iceland basin (regions in right inset) for the c) HR-CESM and d) LR-CESM. The solid (dotted) curves indicate the PI control (Hist/RCP8.5) simulation. Insets: The vertically-averaged (0 – 100 m and 1,000 – 3,000 m) salinity trends (PI control, model years 21 – 100). We did not indicate significant trends for visibility reasons.

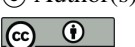



**Figure 4.** The freshwater transport at 34°S (black curve, $F_{ovS}$), 60°N (blue curve, $F_{ovN}$) and the freshwater convergence (red curve, $\Delta F_{ov} = F_{ovS} - F_{ovN}$) for the PI control (upper row) and Hist/RCP8.5 (lower row) simulations. The reanalysis is displayed in the lower row. The yellow shading indicates observed ranges for the $F_{ovS}$.





**Figure 5.** (Upper row): The present-day (1994 – 2020) zonally-averaged meridional velocity at 34°S. (Middle row): The present-day (1994 – 2020) salinity along 34°S. (Lower row): The present-day (1994 – 2020) freshwater transport with depth at 34°S. The present-day profiles originate from reanalysis, and the HR-CESM and LR-CESM under the Hist/RCP8.5 forcing scenario.

## 3.2 The Present-day Comparison

A comparison of the CESM results with reanalysis data over the years 1994 – 2020 (Figure 5) shows that there are large biases in the patterns of ASW, AIW and NADW in the Hist/RCP8.5 simulations. Whereas the meridional velocities at 34°S are reasonably simulated (Figures 5a,b,c), the ASW is too fresh, in particular in the eastern part of the Atlantic and the NADW is too salty (Figures 5d,e,f), as explained in the the previous subsection. The Hist/RCP8.5 simulations consequently have a

positive $F_{\text{ovS}}$ bias upon initialisation and during the years 1994 – 2020 (compare cyan and red curves in Figure 2) .





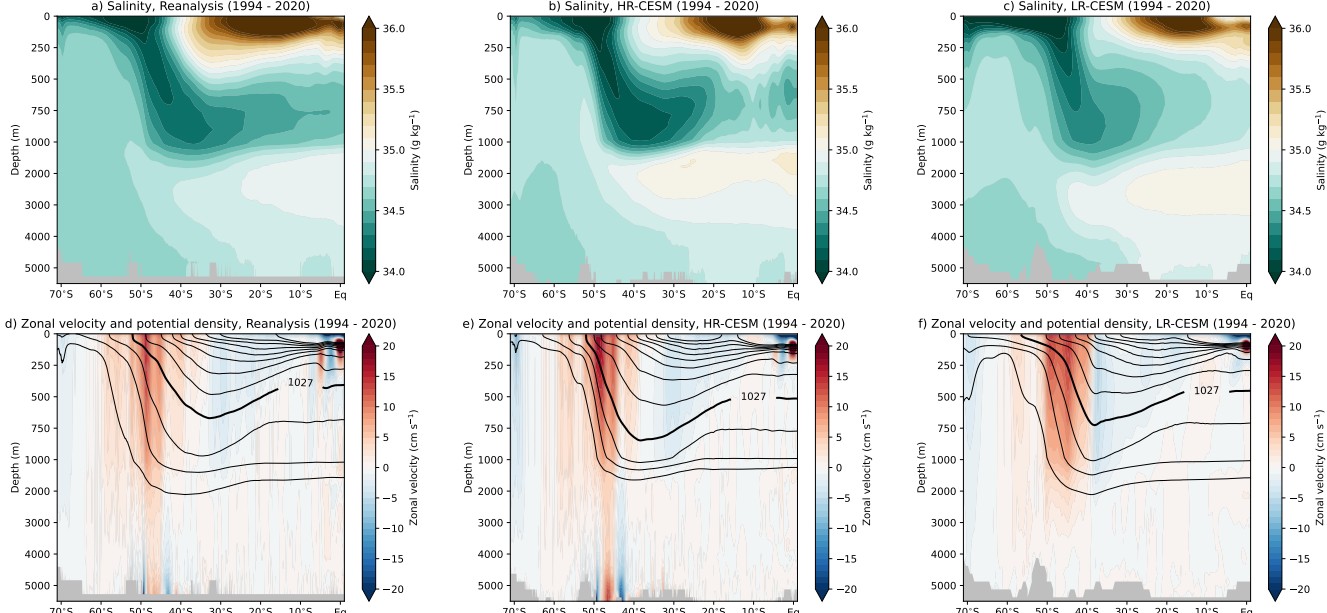

**Figure 6.** (Upper row): The present-day (1994 – 2020) and zonally-averaged (50°W – 20°E, Atlantic sector) salinity. (Lower row): The present-day (1994 – 2020) and zonally-averaged (50°W – 20°E, Atlantic sector) zonal velocity (shading) and potential density (contours are the isopycnals), the contours are each spaced by 0.25 kg m$^{-3}$ and where the thick contour is the 1027 kg m$^{-3}$ for reference. The present-day profiles originate from reanalysis, and the HR-CESM and LR-CESM under the Hist/RCP8.5 forcing scenario.

The AIW originates from the Antarctic Convergence zone (near 50°S – 60°S) and submerges when flowing northward, as shown in Figure 6. In the HR-CESM, the shape (not the absolute values) and outcropping of the isopycnals resembles that of reanalysis and the pattern of the AIW is well represented in the HR-CESM. The zonal velocities, which are related to the Antarctic Circumpolar Current (near 50°S), are slightly higher in the HR-CESM than in reanalysis. The shape and outcropping
of the isopycnals are substantially different in the LR-CESM when comparing those to the reanalysis. The outcropping in the LR-CESM occurs further south giving rise to different water mass properties of the AIW. The ventilation of the AIW is not that well resolved in the LR-CESM and this results in a relatively saline AIW compared to reanalysis and HR-CESM (Figure 5). The relatively saline AIW and the too weak meridional velocities (at 34°S) explain why the AIW bias is larger in the LR-CESM than in the HR-CESM.

The biases in the three water masses ASW, NADW and AIW result in freshwater transport biases at 34°S (Figures 5g,h,i), but the biases in the ASW and NADW are the most dominant and induce a positive $F_{ovS}$ bias. The contribution of the ABW is fairly small and hence we do not discuss it here. The value of $F_{ovN}$ has a small contribution (-0.027 Sv, 1994 – 2020) to the freshwater convergence $\Delta F_{ov}$ in reanalysis. In the HR-CESM, the value of $F_{ovN}$ (-0.030 Sv, 1994 – 2020) is close to reanalysis but for the LR-CESM (-0.080 Sv, 1994 – 2020) it is a factor 3 larger than in the reanalysis (Figure 4). This shows
that $\Delta F_{ov} \approx F_{ovS}(34°S)$ in both the reanalysis and in the HR-CESM.





### 3.3 Climate Change Simulations

The presented quantities in Figures 1, 2, 3 and 4 for the Hist/RCP8.5 simulations remain close to their PI control simulations under the historical forcing (1850 – 2005), but start to deviate in the last 100 years of the simulation. The values of $F_{\text{ovS}}$ decrease under climate change (model years 2000 – 2100, Figures 1a,b) for both the HR-CESM (-0.19 Sv per century, $p < 0.01$) and

LR-CESM (-0.076 Sv per century, $p < 0.01$). Changes in $F_{\text{ovS}}$ can be induced by AMOC changes and/or by salinity changes. The AMOC weakens over the entire Atlantic Ocean and reduces the zonally-averaged meridional velocity magnitudes in the northward flowing branch (upper 1,000 m) and southward flowing branch (1,500 – 4,000 m); as shown in the insets in Figures 7e,f at 34°S. The AMOC strength (Figures 3a,b) decreases by -8.2 Sv per century ($p < 0.01$) and -8.9 Sv per century ($p < 0.01$) for the HR-CESM and LR-CESM simulations, respectively.

The vertically-averaged (0 – 100 m) salinity in the Atlantic Ocean increases under climate change, which is related to negative P-E trends (induced by higher evaporation rates through higher SSTs) over the Atlantic (Figures 7a,b,c,d). Changes in the South American Monsoon result in more precipitation over the South Atlantic Ocean (near 30°S). These changes are the strongest in the LR-CESM leading to a surface freshening around 30°S and 30°W. The upper 100 m salinity over the Indian Ocean decreases by about 0.17 g kg$^{-1}$ per century ($p < 0.05$) for both the HR-CESM and LR-CESM, but there is a south-north

dipole pattern in both salinity and P-E trends. The northward ITCZ shift over the Indian Ocean leads to a different precipitation pattern and results in positive salinity trends in the southern part of the Indian Ocean and, from this, in the Agulhas Leakage (Figures 7e,f). The Indonesian Throughflow shows a freshening of 0.06 Sv per century ($p < 0.1$) and 0.07 Sv per century ($p < 0.01$) for the HR-CESM and LR-CESM (model years 2000 – 2100), respectively (Figure A3).

The salinity response at intermediate depths (250 – 1,000 m) at 34°S is the opposite for the HR-CESM and LR-CESM

(Figures 7e,f) simulations. As discussed above, the outcropping of the isopycnals is different between the HR-CESM and LR-CESM and the outcropping latitude occurs more south in the LR-CESM (somewhere in the center of the Weddell Gyre). The surface waters near the Weddell Gyre therefore show a freshening in the LR-CESM. For the HR-CESM, we find both positive and negative surface salinity trends near the Weddell Gyre. This freshening in the LR-CESM is related to a too strong stratification in the Southern Ocean, which prevents the (deep) vertical mixing of relatively saline water towards the surface

(van Westen and Dijkstra, 2020). The melting of sea ice and snow (on top of the sea ice) contribute to the freshening of the Weddell Gyre in the absence of (deep) vertical mixing in the LR-CESM. The salinity responses below 1,000 m are much smaller and less zonally consistent compared to the upper 1,000 m. The climate change response is delayed at greater depth (by about 100 years, Figure A4), which explains the differences in salinity trends between the upper 1,000 m and those below 1,000 m depth. Salinity changes in the deep water formation regions in the North Atlantic have only a limited effect within this

100-year period (model years 2000 – 2100).

For the HR-CESM, the ASW and AIW are the main contributors (53.9% and 29.5%, respectively) to the $F_{\text{ovS}}$ trend under climate change (Figure 1). The more saline ASW and AIW water masses are the dominant factor in the $F_{\text{ovS}}$ response. The lower zonally-averaged meridional velocities slightly reduce the magnitude of the ASW and AIW trends. For the LR-CESM, the ASW, AIW and NADW contribute 23.7%, 51.3%, 35.4% to the $F_{\text{ovS}}$ trend, respectively (note that the ABW contributes





**Figure 7.** (a & b): The vertically-averaged (0 – 100 m) salinity trends (Hist/RCP8.5, model years 2000 – 2100) for the a) HR-CESM and b) LR-CESM. (c & d): The P-E trends (Hist/RCP8.5, model years 2000 – 2100) for the c) HR-CESM and d) LR-CESM. (e & f): The salinity trends (Hist/RCP8.5, model years 2000 – 2100) along 34°S for the e) HR-CESM and f) LR-CESM. Inset: The zonally-averaged meridional velocity trend at 34°S, the horizontal ranges are between -0.2 and 0.2 cm s$^{-1}$ per century. The hatched regions in all panels indicate significant ($p < 0.05$) trends.





-10.4%). The lower meridional velocities induce the negative ASW and AIW freshwater responses, as these water masses become fresher over time. The negative NADW contribution is related to a freshening of this water mass and this freshening is partly related to changes in the vertical extent of the NADW (it extends into the relatively fresh AIW over time). The AIW, NADW and ABW contributions to the $F_{\mathrm{ovS}}$ trend are 58.6%, 20.6%, -2.9% when fixing the vertical NADW extent to 1,000 – 4,000 m, respectively, the ASW contribution remains unaltered. This effect of the varying NADW extent is smaller in the HR-

CESM (12.8% for varying and 7.4% for fixed NADW). Although the $F_{\mathrm{ovS}}$ decreases in both the HR-CESM and LR-CESM, the $F_{\mathrm{ovS}}$ responses is due to different different processes, where it is mainly salinity dominated in the HR-CESM and overturning dominated in the LR-CESM.

## 3.4   CMIP6 Model Results

The systematic comparison between the HR-CESM and LR-CESM results clearly show the differences in the $F_{\mathrm{ovS}}$ values

and the associated water masses, which are mainly related to the horizontal resolutions between the model configurations. To investigate whether these biases occur also in other models, we include an analysis of $F_{\mathrm{ovS}}$ using 39 different CMIP6 models (under the Hist/SSP5-8.5 scenario). Details about the CMIP6 models used are provided in Table A1. Thirteen out of the 39 CMIP6 models have realistic present-day values of $F_{\mathrm{ovS}}$ (diamonds in Figure 8a) and 7 CMIP6 models have a realistic present-day AMOC strength (circles in Figure 8a). None of the CMIP6 models (and the HR-CESM and LR-CESM) have both

a realistic present-day $F_{\mathrm{ovS}}$ and realistic AMOC strength. The 26 CMIP6 models with a positive $F_{\mathrm{ovS}}$ bias have a stronger AMOC strength compared to the 13 models with a realistic $F_{\mathrm{ovS}}$ (AMOC mean of 17.4 Sv and 12.6 Sv, respectively). Similar to the HR-CESM and LR-CESM, most of the $F_{\mathrm{ovS}}$ bias can be explained by the ASW and NADW contributions (Figures 8c,g).

   For the 13 CMIP6 models with a realistic $F_{\mathrm{ovS}}$, only four of them (CNRM-CM6-1, CNRM-ESM2-1, MCM-UA-1-0 and MRI-ESM2-0) have a reasonable present-day AMOC strength ($\approx 15.5$ Sv), but the remaining ones have a fairly weak AMOC

strength ($< 13.3$ Sv). The CNRM-CM6-1, CNRM-ESM2-1 and MRI-ESM2-0 are relatively fresh (w.r.t. reanalysis) near $10°$W and the surface, which results in a positive freshwater bias for the ASW but this is compensated by a smaller AIW contribution. This relatively freshwater bias appears (to some extent) in most of the CMIP6 models (Figure A6) and also in the HR-CESM and LR-CESM (Figure 5). The displayed CMIP6 profiles in Figure A6 are somewhat small and should only be used for pattern comparison. The freshwater bias near the surface is smaller in the MCM-UA-1-0 and is the one model closest to reanalysis for

the AIW freshwater contribution (Figure 8e). However, for the MCM-UA-1-0 the positive ASW freshwater bias is compensated by a stronger NADW freshwater export out of the Atlantic Ocean. There are 7 CMIP6 models with a strong positive freshwater bias ($F_{\mathrm{ovS}} > 0.2$ Sv) and these models (e.g., FGOALS-f3-L, GISS-E2-2-G, TaiESM1) have an unrealistic mean state at $34°$S. There is only one model (MCM-UA-1-0) which is close to reanalysis for the AIW contribution (Figures 8e), and most models underestimate the AIW contribution.

The MCM-UA-1-0 appears to be the model closest to observations and reanalysis, but this qualification changes when determining the present-day salinity and zonally-averaged meridional velocity root-mean-square errors (RMSEs) w.r.t. reanalysis at $34°$S (inset in Figure 8h). The MCM-UA-1-0 has the second largest salinity RMSE and second largest velocity RMSE of all the diamond-labelled models (i.e., realistic $F_{\mathrm{ovS}}$). The diamond-labelled models have on average the smallest salinity





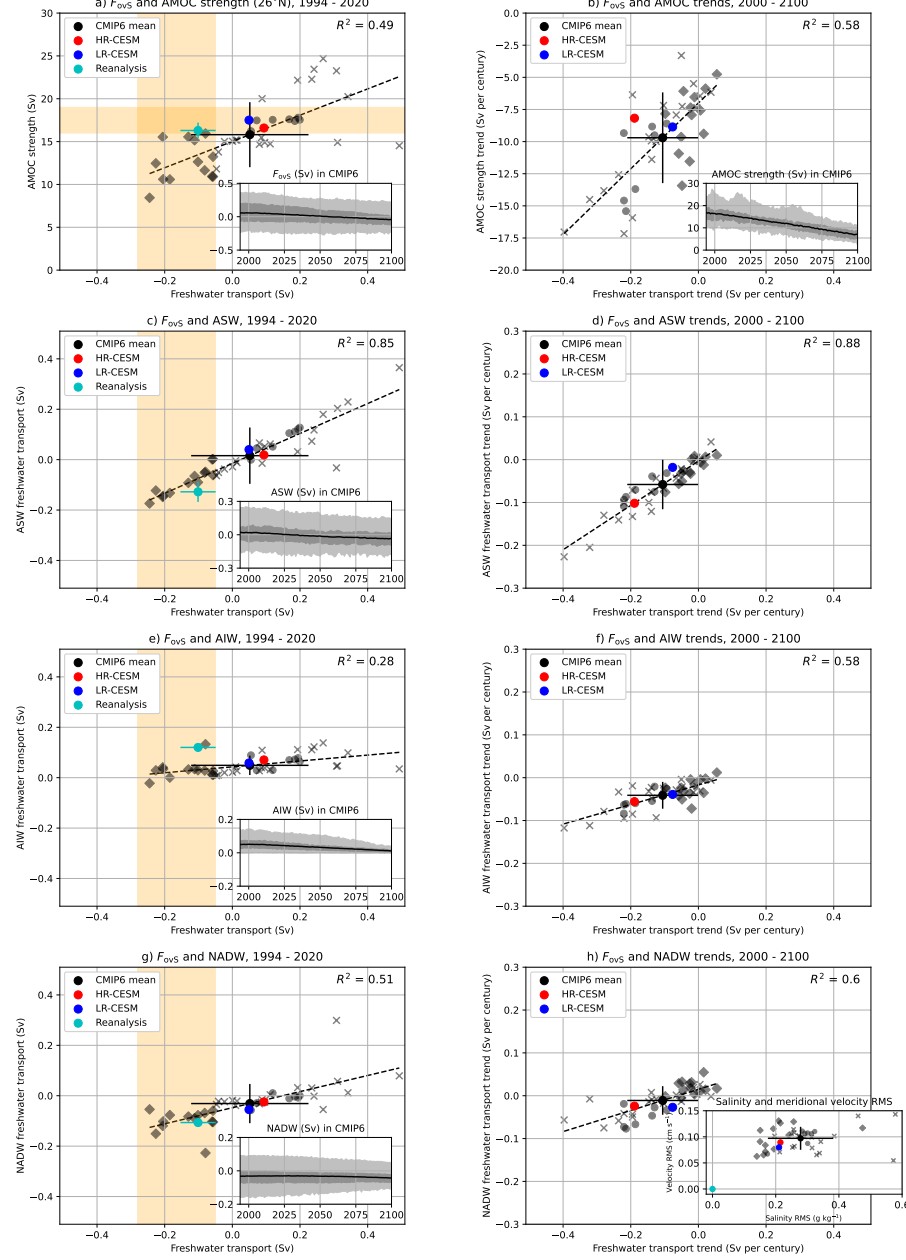

**Figure 8.** Left column: The present-day (model years 1994 – 2020) freshwater transport (at 34°S) and a) AMOC strength (at 26°N and 1,000 m), c) ASW freshwater transport, e) AIW freshwater transport and g) NADW freshwater transport for CMIP6, HR-CESM, LR-CESM and reanalysis. The black diamond and circle markers are CMIP6 models which have a realistic (i.e., within a yellow band) $F_{\mathrm{ovS}}$ and AMOC strength, respectively, whereas the black cross markers fall outside the yellow bands. Right column: Similar to a), c) and e), g), but now the trends (model years 2000 – 2100) in the freshwater transport (components) and AMOC strength. The insets in a), b), c), e), g) show the CMIP6 model mean (black line) and CMIP6 model variance (50% and 95%-confidence levels, shading) for the freshwater transports and AMOC strength over time. The inset in h) shows the model deviations w.r.t. reanalysis for the present-day salinity section and zonally-averaged (baroclinic) meridional velocity profile (at 34°S), here expressed as the weighted root-mean-square errors. The CMIP6 model mean and model standard deviation are also indicated in all panels. The dashed lines in all panels indicate the CMIP6 model regression, the $R^2$ value is indicated in the top right corner.





biases (relatively low salinity RMSE) of the CMIP6 suite, but regarding the velocity RMSE they are not considerably better

than the other CMIP6 models because the diamond-labelled models have a relatively weak AMOC. The FGOALS-g3 has the lowest velocity RMSE, but this model has an unrealistic salinity profile and relatively strong AMOC strength (23.3 Sv) when comparing to reanalysis. These results underline that having a realistic $F_{\mathrm{ovS}}$ does not imply a realistic present-day mean state.

Similar to the CESM results, we find decreasing values in $F_{\mathrm{ovS}}$ (and its components) and AMOC strength under climate change (Figures 8b,d,f,h). The 13 CMIP6 models with a realistic present-day $F_{\mathrm{ovS}}$ show a much smaller $F_{\mathrm{ovS}}$ trend (-0.022 Sv

per century) than in the remaining 26 CMIP6 models (-0.15 Sv per century). The ASW response is the dominant contributor in the $F_{\mathrm{ovS}}$ trend. Note that these $F_{\mathrm{ovS}}$ trends can either be salinity driven (as in the HR-CESM) or overturning driven (as in the LR-CESM).

The HR-CESM and LR-CESM results are consistent with the CMIP6 results and the CESM is actually close to the CMIP6 mean (Figures 8). Most CMIP6 models and the CESM simulations are too fresh near the surface at 34°S (i.e., the ASW

contribution), resulting in a positive freshwater $F_{\mathrm{ovS}}$ bias compared to observations. Models with a realistic $F_{\mathrm{ovS}}$ have either biases in the AIW contribution, NADW contribution or AMOC strength. None of the models analysed here has a realistic present-day mean state when compared to available observations and reanalysis.

### 3.5 Summary and Discussion

Our analysis of CMIP6 models and high-resolution (HR) and low-resolution (LR) versions of the CESM has shown that

persistent biases in these models remain in the AMOC induced Atlantic freshwater transport, as measured by $F_{\mathrm{ovS}}$. The values of $F_{\mathrm{ovS}}$ from the reanalysis product (which is steered towards observations) is in good agreement with those from direct observations (Bryden et al., 2011; Garzoli et al., 2013). In the climate model simulations, numerous processes contribute to this deficiency in $F_{\mathrm{ovS}}$: ITCZ positioning and strength, Agulhas Leakage, Indonesian Throughflow, AMOC strength, and ventilation of the AIW and NADW. Biases in the ASW induce the most dominant $F_{\mathrm{ovS}}$ biases and occur on relative short time

scales (years). Biases in the NADW also induce $F_{\mathrm{ovS}}$ biases but occur on longer (decadal-to-centennial) time scales.

Several model studies (Small et al., 2014; Jüling et al., 2021; van Westen et al., 2020; van Westen and Dijkstra, 2021) demonstrated oceanic bias reductions when increasing the horizontal resolution in the ocean model. However, here the $F_{\mathrm{ovS}}$ bias is larger in the HR-CESM PI control than the LR-CESM PI control (after model year 150). This larger bias is related to a faster (oceanic) adjustment in the higher horizontal resolution model which allows for more eddy-induced horizontal mixing

ventilation. The freshwater convergence/divergence ($\Delta F_{\mathrm{ov}}$) is, however, fairly similar in HR-CESM and LR-CESM, which is related to a relatively large contribution of $F_{\mathrm{ovN}}$ in the LR-CESM. The ASW freshwater transport is fairly similar between the HR-CESM and LR-CESM PI control, but this contribution is mainly related to Indian Ocean's surface (0 – 100 m) salinity and is influenced by precipitation and the Indonesian Throughflow. These results suggest that increasing the ocean model horizontal resolution would have a limited impact on $F_{\mathrm{ovS}}$ biases as these biases are strongly controlled by those in the atmospheric model

component.

To further explore the influence of atmospheric freshwater biases on $F_{\mathrm{ovS}}$, we have conducted simulations with only the ocean component of the CESM (i.e., the Parallel Ocean Program, POP) with the prescribed Coordinated Ocean Reference



Experiment (CORE, derived from observations) forcing dataset (Large and Yeager, 2004; Weijer et al., 2012; Le Bars et al., 2016). The surface (0 – 100 m) salinity biases substantially reduce in the Indian Ocean in the stand-alone POP simulation (0.1° horizontal resolution) and hence reduce the ASW biases (Figure 9). The NADW in the stand-alone POP remains close to reanalysis after 250 years of model integration, whereas the NADW in the HR-CESM PI control simulation has strongly drifted over this period (Figure 2e). This indicates that the atmospheric component and fluxes needs to be improved in the coupled climate simulations to have a realistic salinity distribution, specifically in the Indian Ocean. Once in coupled interaction with the other model components, this would likely then reduce the biases in the Atlantic Surface Water component of $F_{\text{ovS}}$.

The biases in $F_{\text{ovS}}$ due to atmospheric biases is found not only in CESM, but in a large number of CMIP6 models. The CMIP6 model mean has a positive $F_{\text{ovS}}$ bias which is similar as in the CMIP5 results (Mecking et al., 2017). Values of $F_{\text{ovS}}$ decrease under climate change in both versions of the CESM, but the changes are salinity driven in the HR-CESM while for the LR-CESM the changes are overturning driven. Most of the CMIP6 models have similar biases as in the CESM. The models with a realistic $F_{\text{ovS}}$ have biases elsewhere, for example their $F_{\text{ovS}}$ contributions of the AIW and their AMOC strengths are underestimated. The bottom line is that CMIP6 models either have a too weak present-day AMOC or have a wrong sign of $F_{\text{ovS}}$.

In conceptual model, the value of $F_{\text{ovS}}$ is directly related to the strength of the salt-advection feedback. The salt-advection feedback plays a crucial role in AMOC weakening and/or tipping and when this feedback is not well represented (through a too weak AMOC or positive $F_{\text{ovS}}$) the AMOC response is likely to be underestimated. Some studies (Dijkstra, 2007; Huisman et al., 2010) suggest a more versatile role for $F_{\text{ovS}}$ in which the sign of $\Delta F_{\text{ov}} \sim F_{\text{ovS}}$ is also an indicator of whether the AMOC is in a multi-stable regime or not. However, there has been substantial criticism on this aspect of $F_{\text{ovS}}$ (Gent, 2018; Haines et al., 2022). For example, Haines et al. (2022) show that in 10 CMIP5 models the variations in $F_{\text{ovS}}$ do not influence the AMOC strength. However, the AMOC strength in these models poorly matches with that from observations, likely related to a coarse (> 1°) horizontal ocean resolution. In Gent (2018) it is stated that the wind-driven salinity transport is not taken into account property when the AMOC strength varies. However, as argued in Weijer et al. (2019), the wind-driven transport is ineffective in changing the salinity in the Atlantic as a whole and hence does not control the stability of the AMOC. Atmospheric feedbacks, such as the shift of the ITCZ due to AMOC, are not accounted for in $F_{\text{ovS}}$, but the available model studies (Den Toom et al., 2012; Castellana and Dijkstra, 2020) have indicated that these effects are small. While this issue is far from settled, if $F_{\text{ovS}} < 0$ is indeed an indicator for the existence of a multi-stable AMOC regime then models with $F_{\text{ovS}} > 0$ grossly underestimate the probability that an AMOC collapse can occur.

In state-of-the-art climate models, such as in the latest CMIP6 models, the AMOC weakens under future climate change (Weijer et al., 2019; van Westen et al., 2020) but no collapses are found. However, it is questionable whether these climate models are fit for purpose to determine the risk of AMOC tipping, because of their biases identified here. The probability of transitioning to a weakened or collapsed state of the AMOC under climate change is likely underestimated because the major feedback, the salt-advection feedback, is not adequately represented. Because such AMOC weakening and/or tipping can disrupt society worldwide within a few decades, it is very urgent that the model biases are being reduced so that proper estimates of tipping probabilities can be obtained.



**Figure 9.** The present-day (1994 – 2020) and vertically-averaged (0 – 100 m) salinity for a) Reanalysis, b) HR-CESM and c) LR-CESM. For the d) stand-alone POP the time mean of model years 245 – 274 is shown. (e & f): The freshwater transport at 34°S and its components for e) Reanalysis and the f) stand-alone POP, the time series for the HR-CESM and LR-CESM are already shown in Figure 2.



*Code and data availability.* Model output for the CESM simulations can accessed at https://ihesp.github.io/archive/, the processed model output and analysis scripts can be accessed at https://doi.org/10.5281/zenodo.8091845.

The reanalysis product is available at https://doi.org/10.48670/moi-00021. The CMIP6 model output is provided by the World Climate Research Programme's Working Group on Coupled Modeling.

*Author contributions.* R.M.v.W. and H.A.D. conceived the idea for this study. R.M.v.W. conducted the analysis and prepared all figures. Both authors were actively involved in the interpretation of the analysis results and the writing process.

*Competing interests.* The authors declare no competing interests.

*Acknowledgements.* The analysis of all the model output was conducted on the Dutch National Supercomputer Snellius. R.M.v.W. and H.A.D. are funded by the European Research Council through the ERC-AdG project TAOC (project 101055096)





**Figure A1.** The P-E, vertically-averaged (0 – 100 m and 1000 – 3000 m) salinity and SST trends for the LR-CESM PI control simulation. The trends are similar to the insets in Figures 1 and 3, but the trends here are determined over model years 1 – 100.



**Figure A2.** The P-E, precipitation and evaporation trends in the PI control simulations (model years 21 – 100) for the HR-CESM (left column) and LR-CESM (right column). Note the flipped colourbar for the evaporation trends.



**Figure A3.** (a & b) The freshwater transport at $115°E$ (black section in inset) for the a) HR-CESM and b) LR-CESM, where positive (negative) values indicate freshwater transport into (out of) the Indian Ocean. Inset: The vertically-averaged ($0 – 100$ m) salinity trend (PI control, model years $21 – 100$). (c & d): The meridionally-averaged zonal velocity at $115°E$ for the c) HR-CESM and d) LR-CESM PI control simulations (model year 50). (e & f): Salinity along $115°E$ for the e) HR-CESM and f) LR-CESM PI control simulations (model year 50).



**Figure A4.** (a – d): The age of the water (age → 0 at $z = 0$) along $34°$S for model years 50 and 450 for the PI control simulations. The age is presented as the fraction of the maximum age (i.e., 50 and 450 years). (e – h): The spatially-averaged age at $34°$S for the four different water masses for the PI control simulations.





**Figure A5.** The P-E, precipitation and evaporation trends in the PI control simulations (model years 21 − 100) for the HR-CESM (left column) and LR-CESM (right column). Note the flipped colourbar for the evaporation trends.





**Figure A6.** The present-day (1994 – 2020) salinity at 34°S for the 39 CMIP6 models and reanalysis (lower right). The inset shows the freshwater transport with depth at 34°S, the horizontal ranges are between -0.1 and 0.1 mSv m$^{-1}$. The present-day (1994 – 2020) $F_{\mathrm{ovS}}$ and AMOC strength (1,000 m and 26°N) are displayed at the top of each panel. The dashed lines indicate the different water masses (top to bottom: ASW, AIW, NADW and ABW).



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



**Table A1.** The models used in this study with the dimensions of the ocean component, the AMOC strength and the $F_{\mathrm{ovS}}$ (contributions) for the present-day period (1994 – 2020).

| Model name | Number of dimensions (lon × lat × depth) | AMOC (Sv) | $F_{\mathrm{ovS}}$ (Sv) | ASW (Sv) | AIW (Sv) | NADW (Sv) | ABW (Sv) |
|---|---|---|---|---|---|---|---|
| Reanalysis | 4320 × 2041 × 50 | 16.3 | -0.10 | -0.13 | 0.12 | -0.11 | 0.01 |
| HR-CESM | 3600 × 2400 × 62 | 16.6 | 0.09 | 0.02 | 0.07 | -0.02 | 0.03 |
| LR-CESM | 320 × 384 × 60 | 17.5 | 0.05 | 0.04 | 0.06 | -0.05 | 0.01 |
| ACCESS-CM2 | 360 × 300 × 50 | 17.5 | 0.07 | 0.05 | 0.03 | -0.03 | 0.03 |
| ACCESS-ESM1-5 | 360 × 300 × 50 | 17.5 | 0.12 | 0.05 | 0.03 | 0.01 | 0.03 |
| BCC-CSM2-MR | 360 × 232 × 40 | 20.0 | 0.09 | -0.01 | 0.11 | -0.02 | 0.01 |
| CAMS-CSM1-0 | 360 × 200 × 50 | 11.8 | -0.05 | -0.06 | 0.01 | -0.03 | 0.04 |
| CanESM5 | 360 × 291 × 45 | 10.9 | -0.06 | 0.0 | 0.01 | -0.1 | 0.03 |
| CanESM5-CanOE | 360 × 291 × 45 | 10.9 | -0.06 | 0.0 | 0.01 | -0.1 | 0.03 |
| CAS-ESM2-0 | 360 × 196 × 30 | 14.9 | 0.31 | 0.2 | 0.05 | 0.06 | 0.0 |
| CESM2 | 320 × 384 × 60 | 17.4 | 0.18 | 0.11 | 0.07 | -0.0 | 0.0 |
| CESM2-FV2 | 320 × 384 × 60 | 17.5 | 0.2 | 0.13 | 0.06 | 0.0 | 0.01 |
| CESM2-WACCM | 320 × 384 × 60 | 17.6 | 0.17 | 0.11 | 0.07 | -0.01 | 0.0 |
| CIESM | 320 × 384 × 60 | 11.6 | -0.08 | -0.05 | 0.03 | -0.07 | 0.01 |
| CMCC-CM2-SR5 | 362 × 292 × 50 | 15.5 | 0.09 | 0.04 | 0.04 | -0.03 | 0.03 |
| CMCC-ESM2 | 362 × 292 × 50 | 14.9 | 0.1 | 0.05 | 0.04 | -0.03 | 0.03 |
| CNRM-CM6-1 | 362 × 294 × 75 | 15.2 | -0.11 | -0.07 | 0.03 | -0.09 | 0.01 |
| CNRM-CM6-1-HR | 1442 × 1050 × 75 | 12.5 | -0.23 | -0.12 | 0.03 | -0.15 | 0.02 |
| CNRM-ESM2-1 | 362 × 294 × 75 | 15.5 | -0.13 | -0.09 | 0.03 | -0.08 | 0.01 |
| EC-Earth3 | 362 × 292 × 75 | 13.8 | -0.04 | -0.05 | 0.01 | -0.02 | 0.02 |
| EC-Earth3-CC | 362 × 292 × 75 | 15.0 | 0.0 | -0.03 | 0.02 | -0.02 | 0.03 |
| EC-Earth3-Veg | 362 × 292 × 75 | 15.0 | -0.02 | -0.04 | 0.02 | -0.02 | 0.02 |
| EC-Earth3-Veg-LR | 362 × 292 × 75 | 15.2 | 0.01 | -0.01 | 0.03 | -0.02 | 0.02 |
| FGOALS-f3-L | 360 × 218 × 30 | 14.5 | 0.49 | 0.36 | 0.03 | 0.08 | 0.01 |
| FGOALS-g3 | 360 × 218 × 30 | 23.3 | 0.31 | -0.03 | 0.05 | 0.3 | -0.0 |
| FIO-ESM-2-0 | 320 × 384 × 60 | 17.8 | 0.19 | 0.12 | 0.08 | -0.01 | 0.0 |
| GFDL-CM4 | 1440 × 1080 × 35 | 16.2 | 0.06 | -0.0 | 0.09 | -0.06 | 0.02 |
| GISS-E2-1-G | 288 × 180 × 40 | 23.4 | 0.24 | 0.12 | 0.12 | -0.0 | 0.0 |
| GISS-E2-2-G | 288 × 180 × 40 | 24.7 | 0.27 | 0.18 | 0.14 | -0.05 | 0.01 |
| HadGEM3-GC31-LL | 360 × 330 × 75 | 14.7 | 0.11 | 0.06 | 0.03 | 0.0 | 0.01 |
| HadGEM3-GC31-MM | 1440 × 1205 × 75 | 15.2 | 0.01 | -0.0 | 0.04 | -0.04 | 0.01 |
| IPSL-CM6A-LR | 362 × 332 × 75 | 10.6 | -0.18 | -0.13 | 0.0 | -0.08 | 0.02 |
| MCM-UA-1-0 | 192 × 80 × 18 | 15.9 | -0.08 | -0.05 | 0.13 | -0.23 | 0.06 |
| MIROC-ES2L | 360 × 256 × 63 | 10.6 | -0.2 | -0.14 | 0.03 | -0.1 | 0.01 |
| MIROC6 | 360 × 256 × 63 | 12.6 | -0.1 | -0.09 | 0.03 | -0.08 | 0.04 |
| MPI-ESM1-2-HR | 802 × 404 × 40 | 13.2 | -0.06 | -0.06 | 0.03 | -0.06 | 0.04 |
| MRI-ESM2-0 | 360 × 363 × 61 | 15.5 | -0.21 | -0.15 | 0.04 | -0.12 | 0.02 |
| NESM3 | 362 × 292 × 46 | 8.4 | -0.24 | -0.17 | -0.02 | -0.05 | 0.01 |
| NorESM2-LM | 360 × 385 × 70 | 22.3 | 0.23 | 0.07 | 0.11 | 0.03 | 0.02 |
| NorESM2-MM | 360 × 385 × 70 | 22.2 | 0.19 | 0.03 | 0.11 | 0.03 | 0.02 |
| TaiESM1 | 320 × 384 × 60 | 20.2 | 0.34 | 0.23 | 0.1 | 0.01 | 0.0 |
| UKESM1-0-LL | 360 × 330 × 75 | 14.7 | 0.08 | 0.07 | 0.03 | -0.03 | 0.02 |