# Peer review of "Persistent Climate Model Biases in the Atlantic Ocean's Freshwater Transport"

_EGUsphere, 2023_

## Author Comment (AC1)

**MS-No.:** os-2023-1502

**Version:** Revision

**Title:** Persistent Climate Model Biases in the Atlantic Ocean's Freshwater Transport

**Author(s):** René M. van Westen and Henk A. Dijkstra

**Point-by-point reply to reviewer**

October 25, 2023

We thank the reviewer for their careful reading and for the useful comments on the manuscript.

*In this paper the authors study biases in the AMOC stability metric Fov, by analyzing two CESM simulations with different resolutions, as well as a large number of CMIP6 simulations. The authors conclude that the biases that existed in CMIP3 and CMIP5 persist in CMIP6. Furthermore, they point to several biases in the freshwater budget as likely culprits for these biases in Fov.*

*This is a very thorough analysis, and I commend the authors for the work they have done. That said, the depth of the analysis has gone at the expense of the readability of the manuscript; I have to admit –with some embarrassment– that I have not been able to get past the first pages of the Results section, despite several attempts. In my mind, the information density is far too high to make this a comfortable read. To illustrate this point, page 4 alone refers to Fig. 1 (4 panels plus 7 insets), Fig. 2 (8 panels, each with two insets), and 5 figures in Supplemental. The total number of panels + insets covered on page 4 is 70. That is a lot of information to get one's head around in the space of 30 lines.*

*I hope that the authors will reconsider simplifying the paper and improve its readability. The paper can be slowed down significantly, simply by taking more time to develop the material. Not by adding more information, but by more carefully walking the reader through the argumentation following the key results. I understand that it is a challenging task, but the authors should make an effort to boil down the figures to those that are most critical to the storyline. Relegating more figures to Supplemental would be an option, but*

*it only works if they are indeed treated as being of secondary importance, with limited referencing in the main text to avoid distracting from the main storyline. Although the insets might be useful in some cases (after careful study, Fig. 2 started to make sense), in others they are definitely a distraction (Figs. 1, 3). The insets that are critical to the narrative deserve their own figure and should be described and referenced in the proper order.*

**Author's reply:**

Indeed, the information density is quite high, in particular on page 4, but there certainly is room to slow down the pace of the text and to simplify the figures.

**Changes in manuscript:**

We will rewrite and reduce the pace of the manuscript, in particular the result section. We will also strongly reduce the number of insets in the figures and only present the most relevant quantities in each panel. More specifically, we suggest the following changes to the figures in the manuscript:

- Figure 1 – Remove the P-E trends (insets panels a & b) and salinity trends (insets panels c & d). The P-E trend can be explained in the text and the freshening of the Indian Ocean is clearly depicted by the time series.

- Figure 2 – Remove all insets and only mention the relevant results in the text.

- Figure 3 – Remove the P-E trends (insets panels a & b) and replace the salinity trends (insets panels c & d) by only indicating the three different regions (Labrador, Irminger and Iceland basin).

- Figure A1 – Remove from manuscript and explain in text, the results will still be available through Zenodo.

- Figure A2 – Remove from manuscript and explain in text, the results will still be available through Zenodo.

- Figure A3 – Remove from manuscript and explain in text, the results will still be available through Zenodo.

- Figure A4 – Remove from manuscript and explain in text, the results will still be available through Zenodo.

- Figure A5 – Remove from manuscript and explain in text, the results will still be available through Zenodo.

The presented material is then less dense and is expected to improve readability as suggested by the reviewer.

*It is possible that there is simply too much ground to cover for one paper, in which case the authors might consider splitting it up in two companion papers.*

**Author's reply:**

We believe that splitting the story into two parts is not beneficial for our study. To understand the onset of the CESM biases we need to analyse the pre-industrial simulations and to realistically compare the biases against reanalysis we need to analyse the historical simulations as well. Our claim of persistent model biases can not be made by only analysing the CESM and a full CMIP6 comparison is essential. These analyses alone cover 8 out of the 9 main figures, the far majority of the manuscript. Apart from the present-day comparison, the projected freshwater transport trends under climate change (Figures 7 and 8) are also relevant to the manuscript given the importance of $F_{\text{ovS}}$ as discussed in the manuscript. Substantially revising the text and figures, as suggested by the reviewer, is then sufficient to present the results in one manuscript.

**Changes in manuscript:**

We do not follow the suggestion to split the manuscript into two parts, we follow the aforementioned suggestions by reducing the information density of the manuscript.

**Editor comment (not posted online, 17 July 2023)**:

*This manuscript is within Ocean Science aims and scope and thus the review and discussion process can begin. I note that it is common to use the following acronyms for the following Southern Ocean water masses: Antarctic Intermediate water (AAIW) and Antarctic Bottom Water (AABW).*

**Author's reply:**

Yes, we agree with the editor and will change this to AAIW and AABW.

**Changes in manuscript:**

The acronyms will be changed accordingly in the text, figures and analysis software.

---

## Author Comment (AC2)

**MS-No.:** os-2023-1502

**Version:** Revision

**Title:** Persistent Climate Model Biases in the Atlantic Ocean's Freshwater Transport

**Author(s):** René M. van Westen and Henk A. Dijkstra

**Point-by-point reply to reviewer #2**

November 6, 2023

We thank the reviewer for their careful reading and for the useful comments on the manuscript.

*In this manuscript van Westen and Dijkstra take a detailed look into what is causing the model biases in Fov in the Atlantic at 34S. They investigate the bias by looking at the different water masses at 34S and how they change immediately after the model spins up. These responses are compared in both a high (0.1 degree) and low (1 degree) CESM model and later compared to CMIP6 models and changes in future projections. In the CESM models the surface fresh bias can be related to the impact the Indian Ocean has on the Atlantic Surface waters, while slightly deeper the North Atlantic Deep Water biases are related to issues with surface fluxes in the North Atlantic Subpolar gyre. The manuscript furthermore investigates Fov in CMIP6 models and how it changes in future climate projections.*

*Having begun the review of this manuscript after reviewer 1 posted their response I agree with them on the manuscript. The authors have approached the Fov bias issue from a from the perspective of different water masses, which is a very nice and informative way of investigating the model bias. Therefore, I believe this work is of interest to the community. They have also completed and presented a large amount of analysis. However, there is a large amount of information packed densely into one manuscript and it would benefit from streamlining it and/or splitting the manuscript. Similarly with the figures, some panels could be combined instead of having separate panels for the two models allowing a few of the insets to become their own panels, as opposed to small postage stamps.*

*A few smaller points:*

1. *The introduction seems short and could benefit from being expanded, discussion of the usefulness of Fov as an indicator would nice. See Yin and Stouffer 2007 and Mecking et al. 2016 for a discussion on using the divergence around the subtropical gyre. Also, the role of bias correction using flux adjustment (i.e. Liu et al. 2014, Liu et al. 2017, Jackson 2013). The paper Mignac et al. 2019 also worth mentioning.*

   **Author's reply:**

   Yes we agree, those studies are indeed relevant for the manuscript. The relevance of the $F_{\mathrm{ovS}}$ was already discussed in the manuscript (lines 292 – 305), but this can be mentioned in the introduction.

   **Changes in manuscript:**

   We will rewrite and extend the introduction of the manuscript. We will mention the relevant papers and discuss the usefulness of $F_{\mathrm{ovS}}$ as an indicator.

2. *line 80 – see Menary et al. 2020 figure S1 for a comparison between computing own AMOC and AMOC provided by CMIP6 models.*

   **Author's reply:**

   There are small deviations when using the AMOC streamfunction or the meridional velocities to determine the AMOC strength (in their paper at 35°N). The correlation coefficient between the two methods is very high ($r = 0.96$, Menary et al., 2020) and this provides (strong) confidence to use meridional velocities instead of the AMOC streamfunction. Ideally one would like to use the AMOC streamfunction, but not all CMIP6 model provide the AMOC streamfunction as standard output. To include as many CMIP6 models as possible (39 in total), we use meridional velocities to determine AMOC strength.

**Changes in manuscript:**

We will mention and discuss the study of Menary et al. (2020) here.

3. *What are the initial S&T conditions used in this study?*

   **Author's reply:**

   The ocean component was initialised with the January-mean climato-logical (from the World Ocean Atlas) for potential temperature and salinity and from rest (Chang et al., 2020).

   **Changes in manuscript:**

   We will clarify the initialisation of the ocean component (line 53).

4. *How are the freshwater transports computed in Fig.2 for the different water masses?*

   **Author's reply:**

   For each water mass we determine the vertical integral of the freshwater transport with depth between its vertical extent (e.g., see lower row in Figure 5). For example, the contribution of the Atlantic Surface Water (ASW, upper 500 m) is defined as:

   $$F_{\text{ovS}}(\text{ASW}) = -\frac{1}{S_0} \int_{-500}^{0} \left[ \int_{x_W}^{x_E} v^* \mathrm{d}x \right] \left[ \langle S \rangle - S_0 \right] \mathrm{d}z \qquad (1)$$

   where $S_0 = 35$ g kg$^{-1}$ is a reference salinity. The $v^*$ is defined as $v^* = v - \hat{v}$, where $v$ is the meridional velocity and $\hat{v}$ the section spatially-averaged (i.e., full depth) meridional velocity. The quantity $\langle S \rangle$ indicates the zonally-averaged salinity.

**Changes in manuscript:**

We will clarify the water mass contributions in the section 2 (Methods).

5. *There isn't very much mention about Faz in the manuscript despite being defined. Interestingly, looking at the inset in Figure 1b it is clear that Faz also makes a quick adjustment. One thing that is very noticeable in Figure 5 d,e and f is that there is an azonal structure in ASW.*

   **Author's reply:**

   There is indeed room to elaborate on $F_{azS}$ in the manuscript and compare its magnitude against reanalysis.

   **Changes in manuscript:**

   We will add two new panels to Figure 1 to display the $F_{azS}$ time series (now shown as insets in Figures 1a,b) and add the reanalysis time series for comparison. We will change the text accordingly and discuss the $F_{azS}$ results when applicable.

6. *In figure 5 and A6 it would be nice to see the plots as biases as opposed to absolute values.*

   **Author's reply:**

   This is a nice suggestion but we prefer absolute values over anomalies (w.r.t. reanalysis). The AAIW water mass is now clearly depicted in Figure 5 and its origin (Figure 6) is much harder to interpret when showing the figures as anomalies. In Figure A6 only CMIP6 models are shown and without the reference (i.e., reanalysis) it is somewhat difficult to interpret. Moreover, the reanalysis fields need to be interpolated onto each CESM/CMIP6 model grid and this procedure may

give rise to small errors, in particular near the boundaries of the section.

**Changes in manuscript:**

No changes in the manuscript.

7. *There is no mention in the abstract about the future projection results.*

    **Author's reply:**

    It is indeed good to mention these results in the abstract.

    **Changes in manuscript:**

    We will change the text in the abstract accordingly.

*I believe there are several nice results in this manuscript, and I would be happy to provide a more detailed review of this after the above mentioned comments have been considered.*

References

- Yin, J. and Stouffer, R.J., 2007. Comparison of the stability of the Atlantic thermohaline circulation in two coupled atmosphere–ocean general circulation models. Journal of Climate, 20(17), pp.4293-4315.

- Mecking, J.V., Drijfhout, S.S., Jackson, L.C. and Graham, T., 2016. Stable AMOC off state in an eddy-permitting coupled climate model. Climate Dynamics, 47, pp.2455-2470.

- Liu, W., Liu, Z. and Brady, E.C., 2014. Why is the AMOC monostable in coupled general circulation models?. Journal of Climate, 27(6), pp.2427-2443.

- Liu, W., Xie, S.P., Liu, Z. and Zhu, J., 2017. Overlooked possibility of a collapsed Atlantic Meridional Overturning Circulation in warming climate. Science Advances, 3(1), p.e1601666.

- Jackson, L.C., 2013. Shutdown and recovery of the AMOC in a coupled global climate model: the role of the advective feedback. Geophysical Research Letters, 40(6), pp.1182-1188.

- Mignac, D., Ferreira, D. and Haines, K., 2019. Decoupled freshwater transport and meridional overturning in the South Atlantic. Geophysical Research Letters, 46(4), pp.2178-2186.

- Menary, M.B., Robson, J., Allan, R.P., Booth, B.B., Cassou, C., Gastineau, G., Gregory, J., Hodson, D., Jones, C., Mignot, J. and Ringer, M., 2020. Aerosol-forced AMOC changes in CMIP6 historical simulations. Geophysical Research Letters, 47(14), p.e2020GL088166.

- Chang et al. (2020), An Unprecedented Set of High-Resolution Earth System Simulations for Understanding Multiscale Interactions in Climate Variability and Change, JAMES, https://doi.org/10.1029/2020MS002298

---

## Author Response (AR2)

**MS-No.:** os-2023-1502

**Version:** Revision II

**Title:** Persistent Climate Model Biases in the Atlantic Ocean's Freshwater Transport

**Author(s):** René M. van Westen and Henk A. Dijkstra

**Point-by-point reply to reviewer #1**

**February 20, 2024**

We thank the reviewer again for their careful reading and for the useful comments on the revised manuscript. Below the comments are in italic and our response in normal font.

*In this paper the authors study biases in the AMOC stability metric Fov, by analyzing two CESM simulations with different resolutions, as well as a large number of CMIP6 simulations. The authors conclude that the biases that existed in CMIP3 and CMIP5 persist in CMIP6. Furthermore, they point to several biases in the freshwater budget as likely culprits for these biases in Fov.*

*I recognize that the authors have significantly improved the readability of the manuscript by removing many figures, primarily from supplemental information. I commend them for making this effort. I did not find any fault with the analysis, so in principle the paper is publishable, after considering a few minor comments detailed below.*

*That said, despite the reduction in the number of figures, I still found the paper a tedious read, and I'm concerned that this might be detrimental to the impact of the paper. I think that the main culprit is the fact that this paper drowns in details, making it hard to discern any overarching purpose of the paper. In fact, the introduction lacks an explicit goal altogether, missing an opportunity to provide direction for the subsequent analysis. What is the research question that this paper wants to answer? And what analysis will enable us to answer that question? I'm not recommending a major revision of this paper, but it would help the reader if the authors could include some summary statements here and there, as well as introductory statements that explain why a certain analysis is being done, and how it helps to address the*

*main thrust of the paper.*

**Author's reply:** We have made changes in the introduction (also following the comments of the reviewer below) that make the purpose of the paper, and its scientific specific questions, more clear.

*Minor comments:*

1. *l. 26: I think collapses have been found in models with every level of complexity except for eddy-resolving models, but including the eddy-permitting studies of Mecking et al.*

   **Author's reply:**
   Agreed.

   **Changes in manuscript:**
   We mention this now in the revised introduction.

2. *l. 33: salinity → saline*

   **Author's reply:**
   Agreed.

   **Changes in manuscript:**
   Corrected.

3. *l. 63: climatological → climatologies*

   **Author's reply:**
   Agreed.

   **Changes in manuscript:**
   Corrected.

4. *l. 78: I find it hard to reconcile the use of the term 'baroclinic' with the traditional use, even when simply meant to indicate a velocity profile with the barotropic component removed. Removing a section-averaged velocity surely retains a significant barotropic component of the resulting flow field.*

   **Author's reply:**
   The reviewer is correct.

**Changes in manuscript:**
We have rewritten this sentence and dropped the term 'baroclinic', we refer to the definitions of $v^*$ and $\hat{v}$.

5. *l. 106-107, "In this section we focus...": It's probably best to start the section with this sentence, and it wouldn't hurt to be more explicit about the goal of the section.*

   **Author's reply:**
   Agreed.

   **Changes in manuscript:**
   We start the revised section with this sentence and have included the goal of this section.

6. *l. 107: simulation → simulations*

   **Author's reply:**
   Agreed.

   **Changes in manuscript:**
   Corrected.

7. *l. 112: This applies to LR only.*

   **Author's reply:**
   Both the HR-CESM and LR-CESM remain in statistical equilibrium and the natural variability in the HR-CESM is larger than that of the LR-CESM.

   **Changes in manuscript:**
   We explicitly mention the LR-CESM here and that the HR-CESM (PI Control) displays more natural variability.

8. *l. 113: fairy → fairly*

   **Author's reply:**
   Agreed.

   **Changes in manuscript:**
   Corrected.

9. *l. 114, caption of Fig. 1: Aghulas → Agulhas*

   **Author's reply:**
   Agreed.

**Changes in manuscript:**
Corrected.

10. *l. 149: "...both models, the AMOC...": insert 'while'.*

    **Author's reply:**
    Agreed.

    **Changes in manuscript:**
    Suggestion followed.

11. *l. 159, 311: The statement about the horizontal mixing is not obvious to me. On what evidence (or literature) is this based? How does this relate to the use of GM in LR? And is there no role for vertical (or isopycnal) mixing?*

    **Author's reply:**
    This statement was based on the ideal age of the NADW. The NADW has a much larger ventilation rate in the HR-CESM than the LR-CESM. The contribution of vertical mixing is smaller than that of horizontal mixing in the NADW, hence the HR-CESM has a larger ventilation rate as the eddy transport is explicitly resolved. We removed the ideal age results from the Appendix to enhance readability (as suggested by the reviewer in the previous round), but the results are still available through Zenodo.

    **Changes in manuscript:**
    No changes in the manuscript.

12. *l. 182: "...and is related to...": please rephrase.*

    **Author's reply:**
    Agreed.

    **Changes in manuscript:**
    We have rephrased this sentence.

13. *l. 195: I don't quite understand this: Fig. 5 g, h, and i show that the NADW contribution to FovS is consistently small in both the reanalysis and the models.*

    **Author's reply:**
    Note that the depth axis is cropped below 1,000 m depths. The freshwater transport with depth is indeed much smaller in the NADW than

the ASW. However, the vertical extent of the NADW is about 6 times larger than the ASW and hence the NADW contribution to $F_{\mathrm{ovS}}$ is not small. The contributions for each water mass are shown in Figure 2.

**Changes in manuscript:**
We mention now in the caption of Figure 5 (and other figures) that the vertical axis is cropped.

14. *l. 230: more → further*

    **Author's reply:**
    Agreed.

    **Changes in manuscript:**
    Corrected.

15. *l. 274: freshwater → fresh*

    **Author's reply:**
    Agreed.

    **Changes in manuscript:**
    Corrected.

16. *l. 280: Is it relevant that MCM-UA-1-0 seems to have the lowest resolution of all models?*

    **Author's reply:**
    That's indeed a good point and it is interesting that it has the lowest resolution among all CMIP6 models. It is beyond the scope of this paper to individually analyse the CMIP6 models.

    **Changes in manuscript:**
    In the revised manuscript (lines 287 – 288) we mention the low resolution of the MCM-UA-1-0.

17. *l. 324: needs → need*

    **Author's reply:**
    Agreed.

    **Changes in manuscript:**
    Corrected.

18. *l. 335: Do you mean Weijer et al. (2020) here?*

    **Author's reply:**
    Yes, thank you for noticing this.

    **Changes in manuscript:**
    Corrected.

19. *Section 3.4: Does this analysis shed light on why the current generation of climate models cannot have a correct AMOC and FovS at the same time? This would be incredibly useful to know.*

    **Author's reply:**
    Due to the complexity of the effects of the biases on the AMOC and the freshwater balance of the Atlantic, we can only speculate on this based on the results presented in this paper.

    **Changes in manuscript:**
    In the revised section 3.4, we propose a possible reason why the current generation of climate models cannot have a correct AMOC and FovS.

**Point-by-point reply to reviewer #2**

February 20, 2024

We thank the reviewer again for their careful reading and for the useful comments on the revised manuscript. Below the comments are in italic and our response in normal font.

*I commend the authors on greatly improving the manuscript and enhancing the readability. I think this is a very interesting manuscript and deserves publication. However, I still have a few concerns/suggestions:*

1. *Abstract – The manuscript mainly focuses on the results from LR-CESM and HR-CESM and how they relate to the salinity bias at 34S and only the last section of the results talks about CMIP6. The second sentence does not seem correct, the tipping point behaviour is likely caused by freshwater capping and/or warming of high latitudes as opposed to salt-advection feedback, while the idealised studies have shown that positive salt advection feedback is related to whether the AMOC collapse is stable or the AMOC will strengthen again quickly. Also, the abstract makes it seem like the manuscript is exclusively a CMIP6 study. Finally, since the bias at 34S is the main topic of the manuscript it would be useful to be more specific about the biases, i.e. surface too fresh and too salty at depth.*

   **Author's reply:**
   Thank you for these suggestions to improve the abstract of the paper.

   **Changes in manuscript:**
   We have rewritten the abstract following these suggestions.

2. *Line 18 – recent study on future projections and ocean heat transport: Mecking and Drijfhout 2023*

   **Author's reply:**
   That's indeed a relevant study.

   **Changes in manuscript:**
   We have added the reference in the revised manuscript.

3. *Line 24 – The study Lobelle et al. 2020 has estimated that 29-67 years are required to properly detect a weakening of the AMOC.*

**Author's reply:**
Correct, a longer observational record is needed to settle this debate.

**Changes in manuscript:**
We have rewritten this sentence.

4. *Line 28 – Again here I would say that the transitions are not caused by salt advection feedback but rather the stability of them are (or a transition back to an AMOC-on state)*

   **Author's reply:**
   The salt-advection feedback is affecting both the stability of the system and the transition to a collapsed AMOC state.

   **Changes in manuscript:**
   We have rewritten the sentence and now mention both the AMOC stability and the transitions to the collapsed AMOC state.

5. *Line 33 – Note that the sign of Fov is not dependent on the reference salinity, the reference salinity just acts as a scaling, since the section averaged velocity is subtracted at 34S before Fov is computed.*

   **Author's reply:**
   Agreed.

   **Changes in manuscript:**
   No changes in the text needed.

6. *Line 39–45 – Since the salinity bias is talked about in detail it would be worth introducing the bias in more detail since this is described in several papers:*

7. *Line 40–42 – The salinity bias is often corrected using flux adjustment as can clearly be seen in Yin and Stouffer 2007, Jackson 2013, and Liu et al., 2017 while in Mecking et al. 2016 the salinity bias wasn't corrected directly but the bias was lower in this model (i.e. Figure 8c in Mecking et al 2017).*

   **Author's reply:**
   Agreed.

   **Changes in manuscript:**
   We have included these references in the revision.

8. *Line 45–46 – This sentence is very strongly worded and it has not been shown that the positive Fov is the reason for the models not having a tipping AMOC and I see it more as an indicator for the stability of an AMOC off-state as opposed to the model's ability to tip.*

   **Author's reply:**
   Agreed that this is indeed very strongly worded.

   **Changes in manuscript:**
   We have changed the wording.

9. *Line 39–45 – A new multi-model study show, Jackson et al. 2022, shows that there is no clear link between whether the AMOC recovers and Fov, however, in Figure 9 it hints that the models with negative Fov at 34S might be more likely to recover. I think this paper deserves a mention.*

   **Author's reply:**
   Our $F_{ovS}$ values (Figure 8) are consistent with the presented $F_{ovS}$ values from Jackson et al. (2022) (their Figure 9). In Jackson et al. (2022) they show that two models with a slightly negative $F_{ovS}$ ($\approx -0.05$ Sv) have AMOC recovery: EC-Earth3 and MPI-ESM1-2-HR, but that depends very much on the applied forcing. The AMOC in these models may still be in the multiple equilibrium regime but the forcing may not cross the basin boundary, needed to find a transition to a collapsed state. Nevertheless, we agree that it is worth mentioning the study of Jackson et al. (2022) here.

   **Changes in manuscript:**
   We included the results from Jackson et al. (2022) in the revised introduction (lines $51 - 53$) and the EC-Earth3 and MPI-ESM1-2-HR in the revised discussion (lines $352 - 356$).

10. *Line 58 – Mention that it's the ocean component that has 60 vertical levels*

    **Author's reply:**
    Agreed.

    **Changes in manuscript:**
    We mention the ocean component here.

11. *Figure 1c,d – In the reanalysis data for Faz there is a very big spike around 2000 or just before. Given how large this is it should be mentioned. Is there a known explanation for it? If it is caused by salinity can it be seen in other data sets? Also, Figure 5d there is a lot less zonal gradient in salinity in the ASW than in e and f, but this is likely related the freshwater from the Indian Ocean.*

    **Author's reply:**
    There are salinity variations over the upper 500 m at 34°S in the reanalysis data. The western part (with southward surface flow) displays relatively freshwater anomalies and the eastern part (with northward surface flow) displays saline anomalies. This induces a temporarily weaker zonal salinity gradient over the upper 500 m, resulting in a negative $F_{azS}$ anomaly. The available data is relatively short and it is therefore not possible to further investigate this relatively large deviation in $F_{azS}$, but we attribute the $F_{azS}$ spike to inter-annual variability. The differences in panel d with panels e,f are indeed related to the ASW, as is mentioned in the text.

    **Changes in manuscript:**
    In section 3.2, we added a comment on the reanalysis $F_{azS}$ variability before 2000 (lines 192 – 193).

12. *Figure 7e,f – the inset is very narrow, that it took me a while to notice them, I would suggest making them wider*

    **Author's reply:**
    Agreed.

    **Changes in manuscript:**
    The inset is slightly wider in revised Figure 7.

13. *Line 267 – this is similar to Mecking et al. 2017*

    **Author's reply:**
    There are indeed similarities between CMIP6 and CMIP5 (i.e., Mecking et al. (2017)). We refer to Mecking et al. (2017) in the discussion (lines 337 – 338). We didn't include the reference here as we present the CMIP6 results.

    **Changes in manuscript:**
    No changes.

14. *Figure 8 – The figure label says the CMIP6 dots are black but they are actually grey and would be helpful to add the two different symbols for CMIP6 to the legend*

**Author's reply:**
The individual CMIP6 models are indeed indicated in grey. We didn't add the different symbols to the legend to avoid confusion, the markers are clarified in the revised caption.

**Changes in manuscript:**
We have rewritten the caption of Figure 8.

15. *Line 292–294 – Maybe I missed something, but how was it shown that the trends are salinity versus velocity driven?*

**Author's reply:**
The salinity trends over the upper 1,000 m (Figure 7) have a different sign between the HR-CESM and LR-CESM, resulting that the freshwater transport trends are either salinity driven or overturning driven, respectively. This is extensively discussed in the last paragraph of section 3.3.

**Changes in manuscript:**
No changes.

16. *Figure 9 – Panels b,c,d would be easier to see differences if they were plotted as anomalies w.r.t. the Reanalysis*

**Author's reply:**
Ok, agreed.

**Changes in manuscript:**
We now show the biases w.r.t. reanalysis in revised Figure 9.

17. *Figure A1 – The insets are very narrow and difficult to see. This figure would also benefit from plotting as anomalies w.r.t. reanalysis. As stated in the response to the previous reviews, I disagree that much information will be lost in the regridding process since this is a 2D field and not a computation of transport, where more care must be given to conserve the transport in the regridding process. Also, the different water masses can be seen with the dashed lines*

**Author's reply:**
Ok, agreed. We have regridded the reanalysis data and display the

salinity differences w.r.t. reanalysis. We have removed the insets, the contribution of the different water masses are already presented in Figure 8 and Table A1.

**Changes in manuscript:**
We now show the biases w.r.t. reanalysis in revised Figure A1.

18. *Response to Reviewers - Reviewer 2 response 2 – This change has not been implemented into the manuscript*

**Author's reply:**
This concerns the study of Menary et al. (2020). We already followed this suggestion and it was included in the Methods section.

**Changes in manuscript:**
No changes.